# Aryl Hydrocarbon Receptor (AhR)-Mediated Signaling in iPSC-Derived Human Motor Neurons

**DOI:** 10.3390/ph15070828

**Published:** 2022-07-04

**Authors:** Saima Jalil Imran, Barbora Vagaska, Jan Kriska, Miroslava Anderova, Mario Bortolozzi, Gino Gerosa, Patrizia Ferretti, Radim Vrzal

**Affiliations:** 1Department of Cell Biology and Genetics, Faculty of Science, 77147 Olomouc, Czech Republic; 2Stem Cells and Regenerative Medicine Section, UCL Great Ormond Street Institute of Child Health, University College London, London WC1N 1EH, UK; barborav@gmail.com (B.V.); p.ferretti@ucl.ac.uk (P.F.); 3Department of Cardiac, Thoracic, Vascular Sciences and Public Health, University of Padua, 35128 Padua, Italy; gino.gerosa@unipd.it; 4Department of Cellular Neurophysiology, Institute of Experimental Medicine, Czech Academy of Sciences, 14220 Prague, Czech Republic; jan.kriska@iem.cas.cz (J.K.); miroslava.anderova@iem.cas.cz (M.A.); 5Second Faculty of Medicine, Charles University, 15006 Prague, Czech Republic; 6Department of Physics and Astronomy “G. Galilei”, University of Padua, 35131 Padua, Italy; mario.bortolozzi@unipd.it; 7Veneto Institute of Molecular Medicine (VIMM), 35129 Padua, Italy

**Keywords:** hiPSCs, AhR, motor neurons, CYP1A1, CYP1B1, L-Kyn, KA, stem cell model

## Abstract

Exposure to environmental pollutants and endogenous metabolites that induce aryl hydrocarbon receptor (AhR) expression has been suggested to affect cognitive development and, particularly in boys, also motor function. As current knowledge is based on epidemiological and animal studies, in vitro models are needed to better understand the effects of these compounds in the human nervous system at the molecular level. Here, we investigated expression of AhR pathway components and how they are regulated by AhR ligands in human motor neurons. Motor neurons generated from human induced pluripotent stem cells (hiPSCs) were characterized at the molecular level and by electrophysiology. mRNA levels of AhR target genes, CYP1A1 and CYP1B1 (cytochromes P450 1A1/1B1), and AhR signaling components were monitored in hiPSCs and in differentiated neurons following treatment with AhR ligands, 2,3,7,8,-tetrachlodibenzo-p-dioxin (TCDD), L-kynurenine (L-Kyn), and kynurenic acid (KA), by RT-qPCR. Changes in AhR cellular localization and CYP1A1 activity in neurons treated with AhR ligands were also assessed. The neurons we generated express motor neuron-specific markers and are functional. Transcript levels of CYP1B1, AhR nuclear translocators (ARNT1 and ARNT2) and the AhR repressor (AhRR) change with neuronal differentiation, being significantly higher in neurons than hiPSCs. In contrast, CYP1A1 and AhR transcript levels are slightly lower in neurons than in hiPSCs. The response to TCDD treatment differs in hiPSCs and neurons, with only the latter showing significant CYP1A1 up-regulation. In contrast, TCDD slightly up-regulates CYP1B1 mRNA in hiPSCs, but downregulates it in neurons. Comparison of the effects of different AhR ligands on AhR and some of its target genes in neurons shows that L-Kyn and KA, but not TCDD, regulate AhR expression and differently affect CYP1A1 and CYP1B1 expression. Finally, although TCDD does not significantly affect AhR transcript levels, it induces AhR protein translocation to the nucleus and increases CYP1A1 activity. This is in contrast to L-Kyn and KA, which either do not affect or reduce, respectively, CYP1A1 activity. Expression of components of the AhR signaling pathway are regulated with neuronal differentiation and are differently affected by TCDD, suggesting that pluripotent stem cells might be less sensitive to this toxin than neurons. Crucially, AhR signaling is affected differently by TCDD and other AhR ligands in human motor neurons, suggesting that they can provide a valuable tool for assessing the impact of environmental pollutants.

## 1. Introduction

Aryl hydrocarbon receptor (AhR) is a ligand-activated transcription factor, which was originally recognized as a receptor responsive to environmental pollutants such as benzo[a] pyrene, polychlorinated biphenyls (PCBs) or dioxins [1,2,3,4,5,6], hence its alternative name, dioxin receptor. The AhR gene has been highly conserved through evolution and so has its complex regulation of downstream pathways [7].

AhR is a cytosolic protein that, upon ligand binding, translocates to the nucleus and, upon heterodimerization with the AhR nuclear translocator (ARNT), binds to the DNA sequence known as the dioxin- or xenobiotic-responsive element. This triggers the transcription of several genes, many of them involved in the metabolism of xenobiotics, such as cytochrome P450 (CYP) 1A1/1A2/1B1, or involved in cell cycle regulation, cell proliferation, inflammation, and apoptosis [8,9,10,11,12,13]. One of the most potent activators of AhR is 2,3,7,8-tetrachlorodibenzo-p-dioxin (TCDD, dioxin). Even though it is not metabolized, its presence leads to a massive induction of CYPs, particularly CYP1A1/1A2, especially in the liver. In addition to the biotransformation of xenobiotics, CYPs are involved in the metabolism of endogenous molecules, such as fatty acids, hormones, neurotransmitters, steroids, cholesterol, and vitamins [14,15]. Although CYPs are localized predominantly in the liver and intestine, they can be found in almost every tissue of the human body. CYPs can also metabolically activate and detoxify several xenobiotics which enter the brain, as well as play important functions in brain homeostasis and disease. Indeed, a feature of most AhR ligands is their lipophilic nature, which allows them to cross the blood-brain barrier and affect brain physiology [16,17,18,19,20,21,22]. 

Interestingly, it has been recently suggested that neurotoxicants, including AhR ligands, may represent risk factors for ALS (amyotrophic lateral sclerosis), which results in progressive loss of motor neurons in the brain and spinal cord [23,24]. Furthermore, some epidemiological studies link the prenatal exposure of either PCBs or polyaromatic hydrocarbons (PAHs) in heavily polluted areas with reduced fetal growth, motor impairments, and reduced cognitive development [25,26]. Consistent with these observations, animal studies have suggested that TCDD impairs neural development and that its effect on cerebellar granule neurons may result from displacing binding of an endogenous AhR ligand important for their maturation [27,28,29,30,31,32].

Indeed, the existence of endogenous AhR ligand(s) serving distinct physiological functions has been proposed [30,31], with several putative endogenous ligands shown to bind to AhR with variable affinity and potency [33,34]. It has been reported that L-kynurenine (L-Kyn), the first breakdown product in the indoleamine 2,3-dioxygenase (IDO)-dependent tryptophan degradation pathway, and kynurenic acid (KA), another metabolite of tryptophan, activate AhR [35,36].

A number of endogenous compounds have been reported to function as AhR ligands [34,37]. It has been suggested that L-kynurenine (L-Kyn), the first breakdown product in the indoleamine 2,3-dioxygenase (IDO)-dependent tryptophan degradation pathway, and kynurenic acid (KA), another metabolite of tryptophan, activate AhR [35,36], and that KA’s effect on xenobiotic metabolism and promotion of carcinogenesis is AhR-dependent [33].

The neurotoxicity of TCDD and related compounds is a matter of great public health concern [26,30]. Therefore, it is crucial to gain a better understanding of AhR signaling in the human nervous system in order to elucidate molecular mechanisms that may underlie the epidemiological observations and better understand responses to different compounds. Since AhR ligands have been shown to induce CYP1A1/1B1 expression in rodent brains [28,29,30], we hypothesize that this may be the case also in humans. 

Good in vitro models are much needed to study AhR-mediated signaling in human neural cells. We previously investigated whether differentiated neuroblastoma cells into neuron like cells, SH-SY5Y, could provide a relatively simple and useful tool for this purpose [38]. Unfortunately, SH-SY5Y-derived neurons did not express the AhR transcript, even though a slight induction of CYP1A1/1B1 was detected upon TCDD stimulation. Thus, these cells are not suitable for studying AhR-related signaling. Therefore, one of the aims of this study was to establish a more suitable human neuronal model with a focus on motor neurons, given the reported motor deficits and limited information on how this neuronal subtype responds to TCDD exposure. 

Human motor neurons can be generated from pluripotent stem cells [39,40]. To avoid ethical issues raised by the use of human embryonic stem cells, we generated an induced pluripotent stem cell (hiPSCs) line from human fibroblasts and differentiated the hiPSCs into motor neurons. We show that our differentiated neurons do express motor neuron markers and are functional. Importantly, they express AhR, demonstrating their suitability for studying AhR signaling. In addition, our study shows that several components of the AhR signaling pathway are expressed at higher levels in neurons than in hiPSCs, and that the response to TCDD exposure (e.g., CYP1A1 expression) differs in hiPSCs and motor neurons, suggesting a higher sensitivity of the neurons to this toxin. Finally, comparison of the effects of different AhR ligands on motor neurons has shown that they differently affect AhR expression, its translocation to the nucleus, and transcription of CYP1A1/CYP1B1. Together, this study supports the view that hiPSC-derived motor neurons can provide a valuable tool for assessing the impact of different environmental pollutants. 

## 2. Results 

### 2.1. Generation and Characterization of Motor Neurons

First, we characterized the hiPSCs we generated from human neonatal skin fibroblasts. The success of transfection was verified at 48 h (day 2) (Appendix A). Formation of hiPSC colonies was observed by 21 days, (Appendix A). Successful fibroblast reprogramming was confirmed by the expression of the pluripotency markers, Oct4, SEEA-1, TRA-1-81, and TRA-1-60, assessed by immunocytochemistry (Appendix A). We then induced hiPSC neuronal differentiation as summarized in Figure 1 and monitored changes in morphology and protein and gene expression over time by brightfield microscopy, immunofluorescence, RT-Qpcr, and electrophysiology (Figure 1A–D). 

Staining of hiPSC-derived motor neurons for 54 days showed expression of markers of mature neurons and some neural markers, high molecular weight neurofilaments (NF), MAP2A, βIII-tubulin (Tuj1) and polysialylated-neural cell adhesion molecule (PSA-NCAM), NeuN, Hb9, ChAT, Nestin, GFAP, Calretinin, and Olig2 (Figure 1C). Analysis of transcripts in the differentiated cultures by RT-qPCR showed that in addition to a range of neural markers (*βIII-TUBULIN*, *CALRETININ*, *GAP43*, *MAP2*, and *NF*) our cultures expressed markers of immature and mature motor neurons (*OLIG2*, *ChAT*, and *HOXb9*) consistent with successful generation of motor neurons (Figure 1D).

The neuronal identity of the differentiated hiPSCs was further confirmed by assessing their electrophysiological behavior. In total, we analyzed 87 cells by whole patch clamping recording (Figure 2) and monitored them over time (Figure 2A). We identified MAP2^+^ cells with a complex current pattern (*n* = 23; Figure 2B). The cells displayed fast activating outwardly rectifying K^+^ currents (KA), delayed outwardly rectifying K^+^ currents (KDR; Figure 2C) and inwardly rectifying K^+^ currents (KIR; Appendix A). Two of the tested cells expressed voltage-dependent Na^+^ channels, but they were unable to generate action potentials. Furthermore, we identified round shaped MAP2+/βIII-tubulin+ cells with processes that displayed KA and KDR but no KIR (Appendix A). In addition to K^+^ currents, differentiated cells expressed tetrodotoxin (TTX)-sensitive voltage-dependent Na+ channels (*n* = 5 out of five measured cells: Figure 2E,F). The majority of MAP2+/βIII tubulin+ cells (*n* = 33) showed action potentials (Figure 2E–G). 

Together, the fact that the differentiated cells expressed motor markers, showed an outwardly rectifying current pattern, and had TTX-sensitive Na+ channels was consistent with the successful induction of motor neurons.

### 2.2. Monitoring Expression of AhR Signaling Pathway Components in hiPSCs and Neurons

We first assessed whether expression of AhR, the AhR target genes (CYP1A1/CYP1B1), the two forms of the AhR nuclear translocator (ARNT1 and 2), and the AhR repressor (AhRR) was regulated upon neuronal differentiation. As shown in Figure 3A, AhR transcript levels were lower in neurons than in hiPSCs, but both ARNT1 and ARNT2 were greatly up-regulated in neurons, approximately four-fold and twelve-fold higher, respectively. AhRR was also expressed at significantly higher levels, >20-fold with *p* < 0.05 in neurons than in hiPSCs (Figure 3B). Whereas CYP1A1 expression did not appear to change with neuronal differentiation, CYP1B1 was approximately three-fold higher in neurons than in hiPSCs (Figure 3B).

We then assessed whether expression of AhR, CYP1A1/1B1, and AhRR was affected by TCDD treatment in hiPSCs and hiPSC-derived neurons (Figure 3C–E). hiPSCs and neurons were treated for 24 h with increasing concentrations of TCDD (2.5, 5, and 10 nM), or with the DMSO vehicle (0.1% *v*/*v*) as the control. TCDD led to a strong and dose dependent induction of CYP1A1 mRNA in neurons with the fold induction values ranging from 12- to 30-fold at the selected range of TCDD concentrations, which was found to be statistically significant with *p* < 0.05, but to its downregulation in hiPSCs (Figure 3C). In contrast, CYP1B1 was weakly induced in hiPSCs but suppressed in neurons treated with 5 and 10 nM TCDD (Figure 3D). Surprisingly, AhRR expression in neurons was reduced at all the TCDD concentrations tested to half-fold compared with control, whereas in hiPSCs it was down-regulated to half-fold only at 2.5 and 5 nM and was not affected at the higher concentration (Figure 3E). 

### 2.3. Response of AhR-Target Genes to Different Ligands in Neurons

We then assessed changes in AhR and its target CYP1A1 and CYP1B1 gene expression in motor neurons treated for 24 h with different AhR ligands alone or in combination with the AhR antagonist, MNF (3′methoxy-4′nitroflavone) (Figure 4). 

As shown in Figure 4A–C, treatment with TCDD (2,3,7,8,-tetrachlodibenzo-p-dioxin; 1 and 40 nM) up-regulated expression of AhR and CYP1A1, but not of CYP1B1. Only CYP1A1 expression was reduced by MNF. Together with CYP1A1 up-regulation in motor neurons by TCCD, this is consistent with the classical view of its mechanism of action, with CYP1A1 being its main downstream target. 

In comparison with TCDD, we selected two endogenous ligands of AHR reported in the literature. We selected two doses for this treatment based on data reported previously [41]. Treatment with L-Kyn (L-kynurenine; 1 and 40 µM) up-regulated AhR expression only at the higher dose but this effect was not reduced by MNF (Figure 4). CYP1A1 and CYP1B1 transcript levels were differently affected by L-Kyn, with only CYP1B1 being greatly up-regulated approximately 50-fold with *p* > 0.05. Neither the AhR nor the CYP1A1 increases in transcription levels induced by L-Kyn appeared to be reduced by MNF, but increased CYP1B1 expression was induced by combined L-Kyn and MNF treatment in two independent experiments. 

KA (kynurenic acid) treatment (1 and 40 µM) significantly increased AhR expression 17-fold in motor neurons, with the higher dose increasing CYP1B1 over 1000-fold. This up-regulation was reversed in the presence of MNF, an effect not seen in cells treated with TCDD and L-Kyn. In contrast to TCDD and L-Kyn, KA did not affect CYP1A1 expression, but, like L-Kyn, greatly up-regulated CYP1B1, and this effect appeared to be increased by MNF. 

### 2.4. AhR Protein Expression in Neurons following Exposure to AhR Ligand 

We then investigated whether TCDD, L-Kyn, and KA also induced changes in AhR protein distribution and whether they induced its translocation to the nucleus. To this purpose, neurons were stained for AhR after a 90 min treatment with each ligand. This time point selection was based on previous AhR translocation studies in hepatocytes [42]. As shown in Figure 5, AhR staining in the nucleus was increased by all treatments as compared to DMSO controls, with TCDD-treated neurons showing the greater increase. These data confirm that TCDD triggers AhR translocation in human motor neuron cultures.

### 2.5. CYP1A1 Expression and Activity in Neurons in Response to AhR Stimulation 

Finally, we wished to establish whether AhR up-regulation and changes in CYP1A1 mRNA levels following exposure to TCDD, L-Kyn, and KA were reflected by changes in CYP1A activity. Hence, we used the EROD (7-ethoxy-resorufin-O-deethylase) assay to monitor induction of CYP1A1 in neurons treated for 48 h with the three ligands (Figure 6A). Only TCDD treatment was found to induce a significant increase in CYP1A1 activity; no change was observed in either KA- or L-Kyn-treated neurons. 

Immunofluorescence staining for CYP1A1 in neurons treated for either 24 or 48 h with TCDD showed a progressive increase in protein expression, consistent with high activity detected by EROD at the longer time point (Figure 6B).

Together, changes in AhR distribution and CYP1A1 protein expression and activity are consistent with the gene expression changes observed and indicate that in human neurons, TCDD is a much more powerful activator of AhR than KA and L-Kyn, which have been suggested to be endogenous AhR ligands. 

## 3. Discussion

Human in vitro models are much needed to study the impact of environmental pollutants on the nervous system, as there are differences in the response to neuroactive drugs and neurotoxins across species [1,2,4,6]. Here, we have shown for the first time that neurons with the characteristic profile of cholinergic spinal motor neurons generated from hiPSCs express AhR and provide a good human model for investigating AhR signaling.

Human iPSCs and spinal motor neurons show differences in the AhR signaling pathway. We have shown that all components of the AhR signaling pathway we tested, AhR, ARNT1 and ARNT2, CYP1A1, CYP1B1, and AhRR, were expressed both in hiPSCs and motor neurons, but the levels of expression of ARNT1, ARNT2, CYP1B1, and AhRR were much higher in motor neurons than in hiPSCs. In the presence of various concentrations of TCDD (0.25–10 nM), reduction in AhRR levels was consistently observed but was variable and did not reach statistical significance as compared to control. The protein encoded by this gene represses AhR signal transduction by competing with the AhR receptor for binding to ARNT. The lower levels of AhR in hiPSCs and in differentiated neurons could be the cause of the variable level of suppression of AhRR. Expression of transcripts involved in AhR signaling in our hiPSCs is consistent with a recent study in a human embryonic stem cell line, H9 [43]. However, in that study, Teino et al. reported a significant increase in AhR following induction of neural differentiation, which was not observed in our motor neurons, where the levels of AhR expression were fairly similar in the two populations, with a possible trend toward reduction in neurons. This discrepancy is likely due to the different protocols used and stages of differentiation studied. Whereas Teino et al. assessed AhR expression after 7 days of direct neural induction, we examined expression in well differentiated motor neurons. 

Whereas the associated AhR heterodimerization partner ARNT2, as well as ARNT1, were up-regulated after human motor neuron differentiation from hiPSCs, only ARNT2 was up-regulated in neurons differentiated from murine embryonic carcinoma cells P19 [44]. In contrast, ARNT1 was down-regulated, and the authors suggested this to reflect a switch from the more widely distributed ARNT1 to the neural-specific ARNT2 with differentiation [43,44,45]. The discrepancy between our study and Hao et al. (2013) might be due either to species differences, to the use of carcinoma cells versus normal pluripotent stem cells, or to differences in differentiation protocols/identity of neurons generated [44]. Downstream AhR targets in human motor neurons are differentially induced by AhR ligands.

All the ligands studied here were found to induce AhR expression in our motor neuron cultures. This is of interest, because TCDD exposure has been reported to reduce acetylcholine esterase (AChE) activity in an AhR-dependent fashion in cholinergic neurons derived from a human neuroblastoma cell line, SK-N-SH [46]. Hence changes in AChE could provide a readout for future functional analysis of the AhR pathway and assessment of noxious and protective agents in our human motor neurons. 

It is well established that in the liver, TCDD induces CYP1A1 to higher levels compared to CYP1B1, which is the main CYP1 found in brains, both in animals and humans, and is up regulated in the hippocampus following injury [47]. It was therefore unexpected to find that TCDD induced a large increase in the expression of CYP1A1, but not of CYP1B1, in our motor neuron cultures. However, while information on CYP1A1 expression in the human spinal cord is still missing, in the brain this CYP has been found at higher levels in some regions, such as the motor nucleus of the vagus and the substantia nigra, than others, such as the cortex [48]. It is also important to note differences in the effects of L-Kyn and KA treatment on CYP1A1, with a much greater induction of its expression after L-Kyn than KA treatment, which was very modest indeed. Notwithstanding significant transcript up-regulation following L-Kyn treatment, CYP1A1 catalytic activity induction was observed only in TCDD-treated cells, suggesting either different regulation of transcriptional and enzymatic activity or metabolic conversion of L-Kyn and KA next to TCDD. Together, these findings suggest that TCDD toxicity in motor neurons is mainly mediated by CYP1A1.

Although L-Kyn and KA up-regulated expression of AhR comparably to TCDD, unlike TCDD they both greatly increased expression of CYP1B1. This might reflect more physiological roles of these putative endogenous AhR ligands [34,35,36]. It has been reported that in mouse endothelial cells, CYP1B1 is regulated by Wnt/β-catenin signaling, which plays an important role in the maintenance of the blood-brain barrier, whereas CYP1A1 is not Wnt/β-catenin-dependent and is regulated primarily by AhR [49]. On the other hand, studies using a human cerebral endothelial cell line have pointed at an AhR-mediated regulation of CYP1B1 following TCDD exposure [50]. This highlights the importance of the choice of model for unravelling physiological and pathological roles of the AhR pathway.

Reduction in CYP1A1 expression, and possibly CYP1B1, in TCDD-treated cells in the presence of MNF is consistent with an antagonistic action of this compound. However, combined KA and MNF treatment appears to increase CYP1A1 and CYP1B1 expression in KA-treated cells. This will require further investigation as both antagonistic and agonistic effects of MNF have indeed been reported previously and shown to be species dependent [51]. 

While it is tempting to speculate that in our motor neuron model, the effects of TCDD, L-Kyn and KA might be mediated via AhR in the case of CYP1A1, and an AhR-independent pathway, or an AhR pathway regulated via different feedback mechanisms in the case of CYP1B1, further characterization of this very promising model will be needed to properly unravel complex responses to neurotoxicants.

In conclusion, although there is a vast, and at times contradictory, literature on regulation of the AhR signaling pathway and the response to toxicants in several systems, information on the physiological and pathological modulation of this pathway in the human central nervous system is still lacking, and the motor neuron model characterized here will help to fill the gap. 

## 4. Materials and Methods

### 4.1. Materials

All chemicals, reagents, and kits used in this study were purchased from ThermoFisher Scientific (formerly Life Technologies) (Prague, Czech Republic) unless otherwise indicated. The cytokines and growth factors were purchased from PeproTech (Prague, Czech Republic).

Antibodies were procured from Cell Signalling Technology (Danvers, MA, USA) and Abcam (Cambridge, UK). Dimethyl sulfoxide (DMSO), retinoic acid (RA) and laminin were purchased from Sigma-Aldrich (Prague, Czech Republic). 2,3,7,8-tetrachlorodibenzo-p-dioxin (TCDD), L-Kyn and KA were obtained from Ultra Scientific (North Kingstown, RI, USA). Oligonucleotide primers used in qRT-PCR or end-point PCR reactions were synthesized by Generi Biotech (Hradec Kralove, Czech Republic) or by Integrated DNA Technologies (Leuven, Belgium). Light Cycler 480 Probes Master was obtained from Roche Diagnostic Corporation (Intes Bohemia, Czech Republic). All other chemicals were of the highest quality commercially available.

### 4.2. Generation of hiPSCs from Human Fibroblasts

Human dermal fibroblasts (Cat # 06090717) purchased from the European Collection of Cell Cultures (Salisbury, UK) were expanded in fibroblast medium (DMEM with GlutamaxTM, ESC qualified FBS 10%- and 0.1-mM MEM non-essential amino acid solution). Cells were grown in 5% CO_2_ at 37 °C in a humidified chamber for 2 to 4 days before transfection.

For reprogramming the fibroblasts into the iPSCs, we used an Epi 5 Episomal reprogramming kit purchased from ThermoFisher Scientific, which contains Oct4, Sox2, Kfl4, L-Myc, and Lin28 (OKSML) reprogramming vectors necessary for efficient reprogramming. Plasmid expressing GFP were used as transfection controls. Transfection was performed by nucleofection using a 4D-NucleofectorTM System (Lonza, France). Following transfection, cells were plated on Geltrex^®^ matrix coated 6-well plates containing fibroblast medium for 24 h. Thereafter, fibroblast medium supplemented with N2, B27, and basic fibroblast growth factor (bFGF, 100 μg/mL) was added until day 15. From day 15, the medium was replaced by Essential 8 medium according to the manufacturer’s guideline (Epi5 Episomal iPSC Reprogramming Kit User Guide). At day 21, hiPSC-formed colonies were stained for pluripotency markers e.g., Oct4, SEEA-1, TRA-1-60 and TRA-1-81, as described in the Immunostaining of Pluripotency and Neural Differentiation Markers section. 

### 4.3. Neural Differentiation from hiPSCs

Human iPSCs (hiPSCs) were maintained and differentiated according to an established protocol [42,52]. After maintaining the hiPSCs culture in Knockout Serum Replacement (KOSR) medium for 7 days, embryoid bodies (EB) formed, and they were transferred onto the gelatin (0.1%) coated plates. Only highly specific EBs that were showing a tendency towards differentiation were selected for enrichment of neuroepithelial (NE) cells in the next phase by manually scraping off those colonies having neural tube-like structures. These selected EBs were transferred for differentiation in an adherent culture condition from day 7–10 in Neurobasal medium consisting of DMEM/F12, N2 supplement and nonessential amino acids (NEAA) without fibroblast growth factor 2 (FGF2). Neural tubelike rosettes appeared and at day 15 of differentiation, they were detached mechanically and cultured in suspension in hESCs growth medium (consisting of DMEM/F12, KOSR, glutamine, β-mercaptoethanol and NEAA) until day 24. At day 25, medium was replaced with a neural differentiation medium (containing DMEM/F12, N2, NEAA, heparin, cAMP, brain-derived neurotrophic factor, glial cell-derived neurotrophic factor, and insulin-like growth factor 1). Primitive NE cultures were treated with retinoic acid (RA; 100 nM) from day 30 and puromorphamine (PM; 100 ng/mL) was added from day 35. After 3 following days, for motor neuron differentiation, progenitor cells were cultured as a monolayer on laminin and the medium was changed every 3 or 4 days for maturation over 2 to 3 weeks using the half concentration of the RA and PM in the neural differentiation medium. During the differentiation, the expression of selected transcription factors important in neurogenesis was monitored.

### 4.4. RT-qPCR

Total RNA was isolated using TRI Reagent^®^ (Molecular Research Center, Cincinnati, OH, USA). cDNA was synthesized from 1000 ng of total RNA using M-MLV reverse transcriptase (F-572, Finnzymes) at 42 °C for 60 min in the presence of random hexamers (3801, Takara). qRT-PCR was carried out using a Light Cycler 480 II apparatus (Roche Diagnostic Corporation, Prague, Czech Republic). mRNA levels were assessed using primers and a Universal Probes Library (UPL; Roche Diagnostic Corporation, Prague, Czech Republic). Primer sequences are shown in cA,B. The following program was used for monitoring the expression of all genes: an activation step at 95 °C for 10 min was followed by 45 cycles of PCR (denaturation at 95 °C for 10 s; annealing with elongation at 60 °C for 30 s). The measurements were performed in triplicate. Gene expression was normalized using glyceraldehyde-3-phosphate dehydrogenase (GAPDH) as a housekeeping gene. Data were processed by the delta–delta Ct method. 

#### 4.4.1. Immunostaining of Pluripotency and Neural Differentiation Markers

Antibody staining was performed on 4% paraformaldehyde fixed cells as previously described [44]. Pluripotency marker proteins were detected using antibodies listed in Appendix A. Nuclei were counterstained with DAPI. Stained cells were visualized using a Leica SP2 laser scanning spectral confocal microscope. Image collection and analysis were performed using ImageJ software.

#### 4.4.2. Treatment of iPSCs and Differentiated Neurons 

All compounds were solubilized in 10% DMSO diluted with in culture medium, which was used as control medium. 

For gene expression analysis, cells were treated for 24 h with control medium, different concentrations of TCDD (2,3,7,8,-tetrachlodibenzo-p-dioxin; 1, 2.5, 5 and 40 nM), L-Kyn (L-kynurenine; 1 and 40 µM), or KA (kynurenic acid: 1 and 40 µM). 

For AhR protein localization analysis, neurons were treated either with the DMSO vehicle or TCDD (10 nM), L-Kyn (40 nM) or KA (40 nM) for 90 min and double-labelled for AhR and ß3-tubulin or for CYP1A1.

For CYP1A1 activity analysis, neurons were treated with TCDD (10 nM), L- Kyn (40 nM), or KA (40 nM) for 24 or 48 h (as specified in Section 2).

MNF (3′methoxy-4′nitroflavone; 5 µM), an AhR inhibitor, was used in combination with TCDD (10 nm), L-Kyn (40 nM), and KA (40 nM). Treatment was performed for 24 h.

##### Immunofluorescence 

Antibody (Appendix A) staining was performed on 4% paraformaldehyde fixed cells as previously described [53]. Nuclei were counterstained with DAPI. Stained cells were visualized using a Leica SP2 laser scanning spectral confocal microscope with a 20X objective. Image collection and analysis were performed using ImageJ software.

##### CYP1A1 Enzyme Activity Assay (EROD ASSAY)

7-Ethoxyresorufin-O-deethylation (EROD) activity was determined in live neurons grown in 24-well plates. Cells were washed twice with PBS and then incubated with 100 µL of the PBS containing 8 µM 7-ethoxyresorufin and 10 µM dicumarol to prevent further metabolism of resorufin. After 30 min of incubation at 37 °C, 75 µL was transferred to a 24-well plate together with 125 µL of methanol. The fluorescence of resorufin was measured at 530 nm excitation and 590 nm emission wavelengths using an Infinite M200 machine (Tecan, Männedorf, Switzerland).

### 4.5. Patch-Clamp Recording

Cells were maintained on glass or vinyl (both P-Lab, Prague, Czech Republic) coverslips coated with Geltrex^®^ (Life Technologies, Waltham, MA, USA) at a cell density of 6 × 10^4^/cm^2^.

Cell membrane currents were recorded 15–20 days after plating using the patch-clamp technique in the whole-cell configuration. Recording pipettes with a tip resistance of 8–12 MΩ were made from borosilicate capillaries (Sutter Instruments, Novato, CA, USA) using a P-97 Brown-Flaming micropipette puller (Sutter Instruments, Novato, CA, USA). Recording pipettes were filled with intracellular solution containing (in mM): 130 KCl, 0.5 CaCl_2_, 2 MgCl_2_, 5 EGTA, 3 ATP, and 10 HEPES (pH 7.2). All recordings were made in HEPES-based artificial cerebrospinal fluid (aCSF) containing (in mM): 135 NaCl, 2.7 KCl, 2.5 CaCl_2_, 1 MgCl_2_, 1 Na_2_HPO_4_, 10 glucose, and 10 HEPES (osmolality 312.5 ± 2.5 mmol/kg, pH 7.4). All recordings were made on cover slips perfused with aCSF at room temperature. Electrophysiological data were measured at a 10 kHz sample frequency using an EPC9 amplifier controlled by PatchMaster software (HEKA Elektronik, Lambrecht/Pfalz, Germany) and were filtered using a Bessel filter. The coverslips with cells were transferred to the recording chamber of an upright Axioscop microscope (Zeiss, Gottingen, Germany) equipped with electronic micromanipulators (Luigs &Neumann, Ratingen, Germany) and a high-resolution AxioCam HR digital camera (Zeiss, Gottingen, Germany).

The resting membrane potential (Vm) was measured by switching the EPC9 amplifier to the current-clamp mode. Using FitMaster software (HEKA Elektronik, Lambrecht/Pfalz, Germany), the membrane resistance (IR) was calculated from the current value 40 ms after the onset of the depolarizing 10 mV pulse from the holding potential of −70 mV to −60 mV for 50 ms. Membrane capacitance (Cm) was determined automatically from the Lock-in protocol by PatchMaster. Current patterns were obtained by hyper- and depolarizing the cell membrane from the holding potential of −70 mV to values ranging from −160 mV to 40 mV at 10 mV intervals. Pulse duration was 50 ms. In order to isolate voltage-gated delayed outwardly rectifying K^+^ (KDR) and inwardly rectifying K^+^ (KIR) current components, a voltage step from −70 to −60 mV was used to subtract the time- and voltage-independent passive currents as described previously [54,55]. To activate only KDR currents, the cells were held at −50 mV and the amplitude of the KDR current was measured at 40 mV at the end of the pulse.

The amplitudes of KIR currents were measured at −160 mV at the end of the pulse. The fast activating and inactivating outwardly rectifying K^+^ (KA) current component was isolated by subtracting the current traces clamped at −110 mV from those clamped at −50 mV, and its amplitude was measured at the peak value. Na+ currents were activated by depolarizing steps from −70 to 20 mV and the current component was isolated by subtracting the current traces clamped at the voltage with maximal current activation, and its amplitude was measured at the peak value. To block tetrodotoxin-sensitive voltage-gated Na^+^ channels, 1 μM tetrodotoxin (TTX; Alomone Labs, Jerusalem, Israel) was used. Action potentials were obtained in the current-clamp mode by injecting current. Current values ranged from 50 pA to 1 nA at 50 pA intervals. Pulse duration was 300 ms. After recording, the immunocytochemistry was used to confirm the presence of MAP2 and βIII-tubulin.

### 4.6. Statistical Analysis

All measurements were performed with replicates (*n* ≥ 3 unless otherwise indicated). Dose- and time-course studies of EROD activity and CYP1A1 expression were performed at least in two independent experiments. The tests of the overall null hypotheses were performed using one-way ANOVA tests and followed by student t tests for comparisons of two groups. Graphs were drawn using GraphPad Prism 6 software as means ± standard errors of the mean (SEM), while the level of statistical significance was marked as follows: * *p* < 0.05, ** *p* < 0.01, and *** *p* < 0.001.

## 5. Conclusions

In conclusion, our study, together with work discussed above, supports the view that hiPSC-derived neurons may serve as a valuable human model for assessing AhR activity in relation to developmental and physiological changes induced by dioxin or dioxin-like compounds, which are present in the majority of industrial products, and to their potential impact on human health.

This initial analysis of the expression of various AhR ligand-induced enzymes in human motor neurons suggests a relevant relationship between effects observed in vivo and in vitro following exposure to these toxicants. Future use of this human model for detailed mechanistic studies will extend our understanding of responses mediated by AhR in health and disease.

## Figures and Tables

**Figure 1 pharmaceuticals-15-00828-f001:**
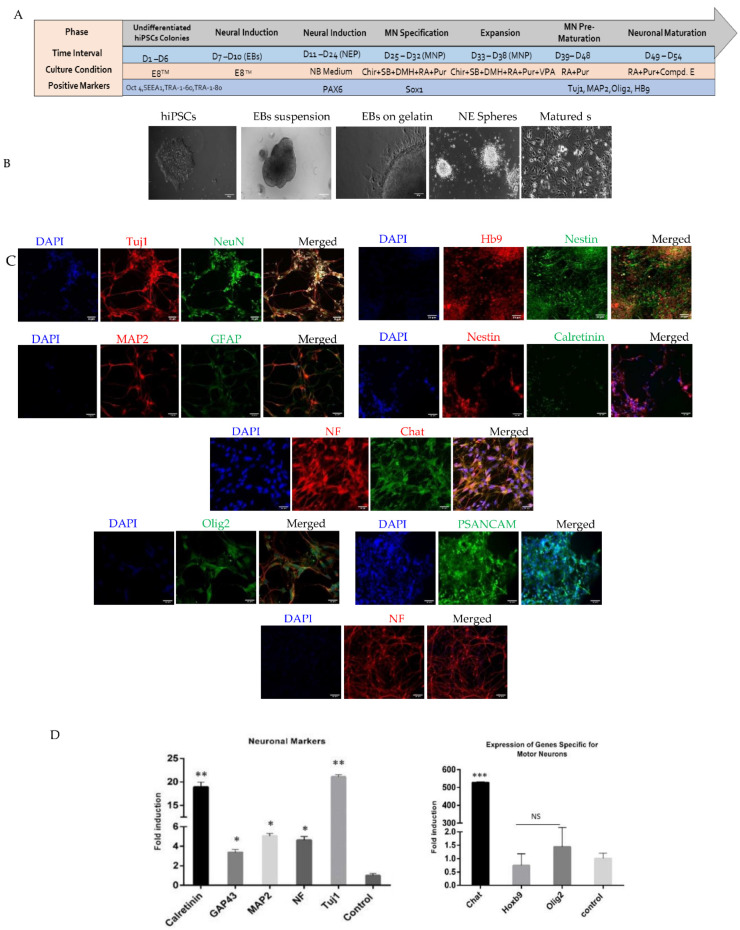
(**A**) Schematic representation of the differentiation strategy for motor neurons using small molecules. (**B**) Phase contrast images of the various phases of differentiation from hiPSC monolayer colonies. Neurospheres, neural tube formation and final maturation phase of motor neurons. Scale bar = 100 μM. (**C**) immunofluorescence data for the expression of general and specific motor neurons merged images are presented with DAPI nuclear staining. All the images were obtained during maturation phase. Scale bar = 50 μM. (**D**) RT·PCR data representing the gene expression for specific and generalized in the motor neuronal culture. Fold induction of gene expression is compared with the control undifferentiated hiPSCs. *n* ≥ 3; * *p* < 0.05. ** *p* < 0.01. *** *p* < 0.001 NS indicates non-significant values.

**Figure 2 pharmaceuticals-15-00828-f002:**
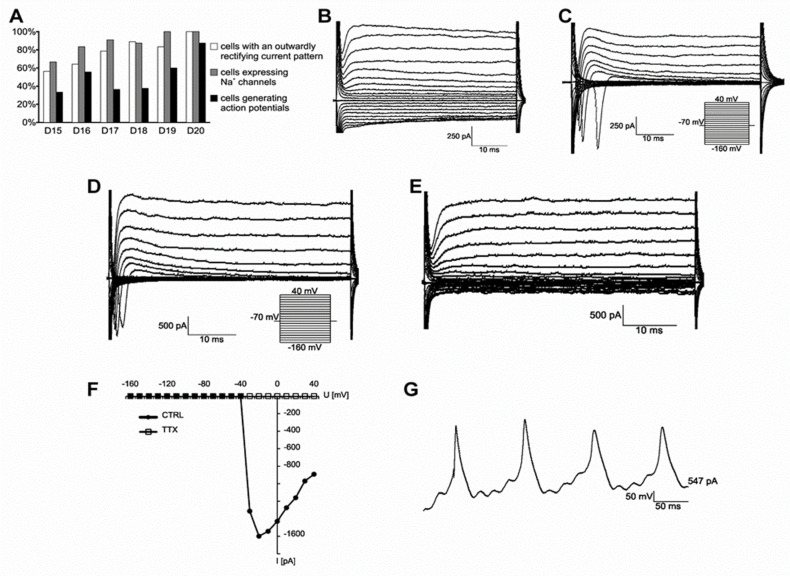
Electrophysiological properties of hiPSC-derived neurons. (**A**) Incidence of cells with an outwardly rectifying current pattern (white columns), cells expressing Na^+^ channels (grey columns), and cells generating action potentials (black columns) during their in vitro differentiation from day 15 (D15) till day 20 (D20). Together, 87 cells were measured, comprising 23 cells with a complex current pattern and 64 cells expressing an outwardly rectifying current pattern. Total numbers of cells (*n*) on respective measuring days: D15 (16), D16 (28), D17 (14), D18 (9), D19 (12) and D20 (8). (**B**) Differentiated cell displaying a complex current pattern. (**C**) Differentiated cell displaying an outwardly rectifying current pattern. (**D**,**E**) Electrophysiological properties of the cells expressing an outwardly rectifying current profile. Representative current pattern prior to (CTRL; (**D**)) and after tetrodotoxin (TTX; (**E**)) application. (**F**) The resulting I/V relationship of Na^+^ current prior to (filled circles) and after TTX (empty squares) application. (**G**) The cells displaying outwardly rectifying K^+^ channels express TTX-sensitive Na^+^ current and are able to generate action potentials in response to current injection. Current patterns were obtained by hyper- and depolarizing the cell membrane from the holding potential of −70 mV to values ranging from −160 mV to 40 mV at 10 mV intervals. Pulse duration was 50 ms. Action potentials were obtained in the current-clamp mode by injecting current. Current values ranged from 50 pA to 1 nA at 50 pA intervals. Pulse duration was 300 ms.

**Figure 3 pharmaceuticals-15-00828-f003:**
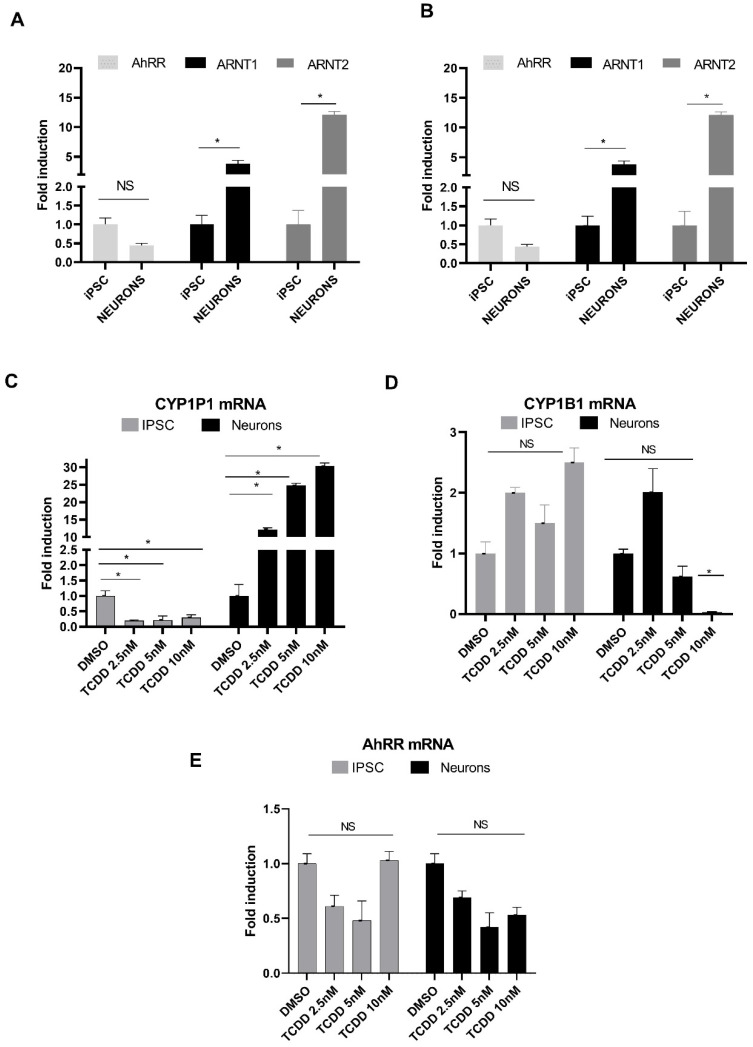
Changes in the expression of components of the AhR signaling pathway following differentiation of iPSCs into motor neurons (**A**,**B**) and in response to 24-h TCDD treatment (**C**–**E**) assessed by RT-qPCR. (**A**,**B**) Note significant higher mRNA expression of ARNT1 and 2, CYP1B1 and AhRR in neurons as compared to iPSCs. (**C**) TCDD treatment increases CYP1A1 expression in neurons but not in iPSCs at all concentrations tested. (**D**) TCDD treatment increases CYP1B1 expression more in iPSCs than in neurons. (**E**) AhRR expression is downregulated by TCDD. *n* ≥ 3; * *p* < 0.05. NS indicates non-significant values.

**Figure 4 pharmaceuticals-15-00828-f004:**
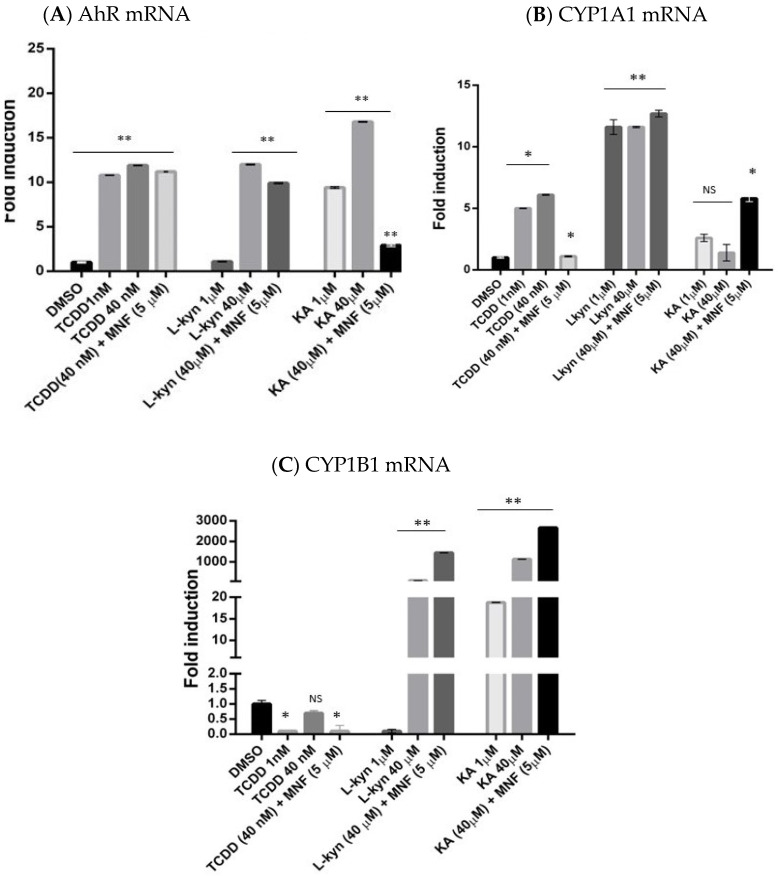
Comparison of the effect of different AhR ligands on AhR expression in motor neurons. Transcription levels after 24-h treatment with TCDD (1 and 40 nM), L-Kyn (1 and 40 µM), KA (1 and 40 µM) alone or in combination with the antagonist, and MNF (5 µM), as assessed by RT-qPCR; neurons treated only with the DMSO vehicle serve as controls. *n* ≥ 3; *p* < 0.05. (**A**) AhR mRNA (**B**) CYP1A1 mRNA, (**C**) CYP1B1 mRNA. (*n* ≥ 3). * *p* < 0.05 ** *p* < 0.01. NS indicates non-significant values.

**Figure 5 pharmaceuticals-15-00828-f005:**
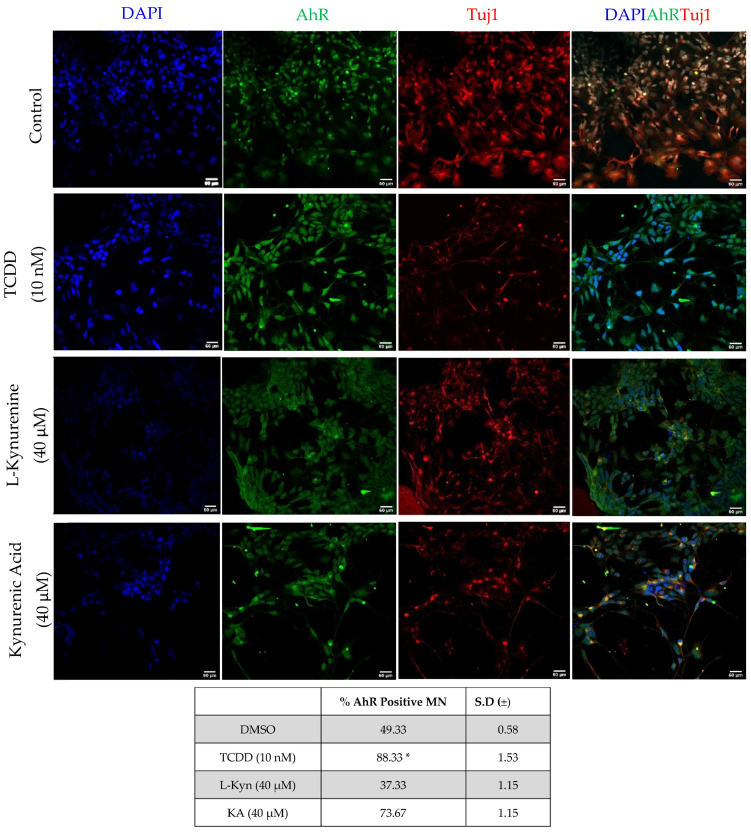
AhR protein distribution in motor neurons in response to treatment with different AhR ligands assessed by immunofluorescence. Neurons were treated for 90 min with TCDD (10 nM)*,* L-Kyn (40 µM) or KA (40 µM) prior to fixation. Note that the most intense AhR staining (green) is observed in TCDD-treated neurons. Tuj1 (Red) and DAPI (Blue) used as reference marker for neurons. Scale bar = 50 μM. ImageJ-assisted quantification of the cells induced to express AhR is shown in the bottom table vurses the DMSO alone controls (*n* ≥ 3). * *p* < 0.05.

**Figure 6 pharmaceuticals-15-00828-f006:**
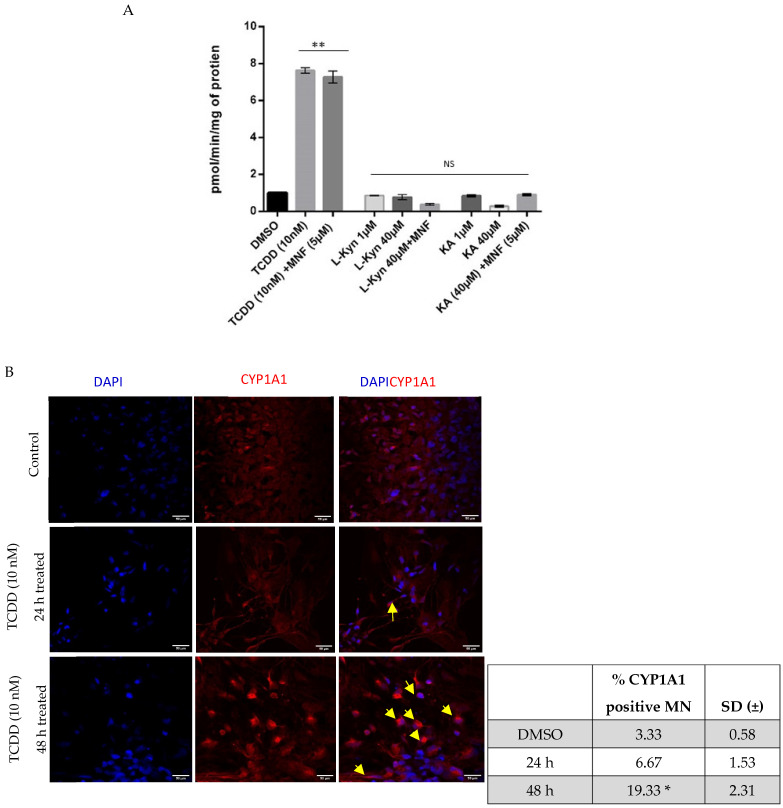
CYP1A1 protein detection in mature neurons. (**A**) CYP1A1 enzyme activity detected by EROD after treatment for 48 h with TCDD, L-Kyn or KA alone or with the AhR antagonist, MNF, at the concentration indicated in the graph. (**B**) Immunofluorescence detection of CYP1A1 by immunofluorescence (red) after treatment with TCDD (10 nM) for either 24 or 48 h. Nuclei are stained with DAPI (blue) (*n* ≥ 3). ** *p* < 0.01. NS indicates non-significant values. CYP1A1-positive cells are indicated by yellow-colored arrows. Scale bar = 50 μM. ImageJ-assisted quantification of the cells induced to express CYP1A1 is shown in the bottom table versus the experimental conditions. (*n* ≥ 3). * *p* < 0.05.

## Data Availability

Data is contained within the article and Appendix A.

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
