# Peer review of "Aryl Hydrocarbon Receptor (AhR)-Mediated Signaling in iPSC-Derived Human Motor Neurons"

_pharmaceuticals, 2022, doi:10.3390/ph15070828_

Round 1

Reviewer 1 Report

In this paper, Imran and colleagues provide an human model of cholinergic spinal motor neurons generated from hiPSC to investigate AhR signaling. Data reported in this manuscript are interesting, however many weaknesses preclude its publication in the present form.

 The following points need to be addressed to further support the authors' conclusions.

- Authors differentiated hiPSC into motor neurons, they should better characterized the profile of cholinergic spinal motor neurons by immunocytochemistry using other motor neuron markers (e.g ChAT);

 -        In figure 1D authors reported Real time data for specific and generalized neuronal markers. Please specify with respect to is calculated the Fold induction (with respect to IPSC ?)

- Authors demonstrated that TCDD treatment led to a strong and dose dependent induction of CYP1A1 mRNA in neurons and its downregulation in hiPSCs. Please provide the fold induction in the text. Please add also: i) the value of the downregulation of AhRR expression in neurons and in iPSC at all the TCDD concentrations tested; ii) p values in the text and in the bar graphs.

-          The authors should discuss better why 10nM of TDDC does not affect AhRR mRNA.

-          As shown in Figures 4A-C, neurons were treated with TCDD at the concentrations of 1 and 40 nM. Why authors changed the concentration of TCDD? Please provide the same data described in fig 3 C-E with 1 and 40nM of TCDD.

-          Treatment with L-Kyn, greatly up-regulated CYP1B1, and this effect appeared to be increased by MNF. Please provide fold induction in the text and p value

-          In the Fig. 6, authors reported an increment of AhR staining in nucleus after all treatments as compared to DMSO. To further support this data, authors should provide a quantification of the nuclear translocation in all conditions reporting mean values and deviation standard of data.

-          In the fig 5, Immunofluorescence staining for CYP1A1 in neurons treated for either 24 or 48 hours with 10 nM of TCDD, showed progressive increase in protein expression, consistent with high activity. To support this data please provide a quantification of immunofluorescence staining for CYP1A1 reporting mean values and deviation standard of data. To support the increase of CYP1A1 expression, western blot experiment would be more appropriate.

-          Values (means and errors) of all experiments must be indicated in the main text;

Minor comments:

Please report scale bars on all immunofluorescence images (in Figs 1B, 5, 6 and Fig. S1). Moreover, technical details of image acquisition (objective used, magnification, image resolution in terms of pixels) are lacking and should be added.

Figs. 1E and 1G are mentioned in the text but are non reported in the figure, please check them;

- At line 216, authors indicated Fig 6B. Please check if they intended Figure 5

-       -    In the figure legend of Fig 5 please specify what the arrows indicate

-          Figs 2D-G and 5 have no reference in the text;

-          Author should justify the concentration of TCDD ( 2.5, 5 and 10 nM) of L-Kyn ( 1 and 40μM) and of KA Kynurenic Acid (1 and 40μM)

-          Texts contain typos: neurones” rather than “neurons”;

-          Error bars are not present in all bar graphs, please check them (e.g Fig 1D, 4 B and C);

-          Please indicate the values of p for each asterisks in the figure legend

-          Please indicate the “not significance” or the p value in all bar graphs.

-          Please specify the software used for statistical analysis;

-   In the materials and methods please indicate:  the secondary antibodies used for Immunofluorescence and the concentration of primary and secondary antibodies. In the paragraph, “Neural differentation from “hIPSC” please reported the final concentration of all components.

Author Response

Dear Prof.

Authors are thankful to the esteemed reviewers for their valuable suggestions which will definitely improved the quality of the text and authenticity of the data and manuscript in general.

All the sections of manuscript have been revised substantially to make them more reflective and easy to understand.

The manuscript has also been revised for grammar and English in consultation with a linguistic expert.

To the best of our knowledge the manuscript is technically and scientifically correct and authentic for publication as a full length research article in your esteemed journal, Pharmaceuticals

Responses to  Reviewer 1

In this paper, Imran and colleagues provide an human model of cholinergic spinal motor neurons generated from hiPSC to investigate AhR signaling. Data reported in this manuscript are interesting, however many weaknesses preclude its publication in the present form.

 The following points need to be addressed to further support the authors' conclusions.

Comment: Authors differentiated hiPSC into motor neurons, they should better characterized the profile of cholinergic spinal motor neurons by immunocytochemistry using other motor neuron markers (e.g ChAT);

Response: Thanks for this suggestion. We have incorporated an image showing staining for the motor neuron marker ChAT in Figure 1 (panel 1C).

Comment: In figure 1D authors reported Real time data for specific and generalized neuronal markers. Please specify with respect to is calculated the Fold induction (with respect to IPSC ?)

Response: In figure 1D, the fold changes shown are in relation to undifferentiated hiPSCs

Comment:  Authors demonstrated that TCDD treatment led to a strong and dose dependent induction of CYP1A1 mRNA in neurons and its downregulation in hiPSCs. Please provide the fold induction in the text. Please add also: i) the value of the downregulation of AhRR expression in neurons and in iPSC at all the TCDD concentrations tested; ii) p values in the text and in the bar graphs.

Response: Thanks for this comment. The fold changes and p values for all genes studied in text and graphs are now indicated.

      Comment: The authors should discuss better why 10nM of TDDC does not affect AhRR mRNA.

      Response: A comment on the variation in AHRR levels at different TCDD concentrations has been included in the Discussion section.

      Comment:  As shown in Figures 4A-C, neurons were treated with TCDD at the concentrations of 1 and 40 nM. Why authors changed the concentration of TCDD? Please provide the same data described in fig 3 C-E with 1 and 40nM of TCDD.

 Response: Thank you for this comment. The purpose of the experiment shown in Fig 3C was to compare responses to TCDD in hiPScs and neurons, and we used concentration known to be effective from pilot experiments. It should be noted that one of the TCDD concentrations, 1 nM, has been used in all experiments. In the set of experiments in Fig. 4 and 5, we were focusing more on the comparison of the effect of different ligands and inhibition of their effects on transcription of components of the AhR pathway. L-Kyn and KA are known to be weaker AhR ligands, hence they were used at higher concentrations in line with other studies. Concerning TCDD, we used one of the concentrations used in the first experiment (the lowest one) and one that was 40 fold higher, to match the fold difference used for the other ligands, L-Kyn and KA. It would take a much longer time than allowed for this review to set an experiment to compare the effect of 40 nM TCDD on hiPSCs and neurons, given the long timeframe for generating motor neurons, and we do not believe such experiment would change the main message emerging from this part of the study, which highlights that changes in responses to TCDD occur with neuronal differentiation, and suggests differences in vulnerability to TCDD in hiPSCs and neurons.

Comment:  Treatment with L-Kyn, greatly up-regulated CYP1B1, and this effect appeared to be increased by MNF. Please provide fold induction in the text and p value

 Response: Thanks for pointing this out. The fold induction and p value have been mentioned in the text.

      Comment:  In the Fig. 6, authors reported an increment of AhR staining in nucleus after all treatments as compared to DMSO. To further support this data, authors should provide a quantification of the nuclear translocation in all conditions reporting mean values and deviation standard of data.

Response: Thank you so much for this useful suggestion. ImageJ assisted quantification of AhR staining in neurons has been calculated and included in a Table in Figure 6 with the mean value and standard deviation indicated.

 Comment:   In the fig 5, Immunofluorescence staining for CYP1A1 in neurons treated for either 24 or 48 hours with 10 nM of TCDD, showed progressive increase in protein expression, consistent with high activity. To support this data please provide a quantification of immunofluorescence staining for CYP1A1 reporting mean values and deviation standard of data. To support the increase of CYP1A1 expression, western blot experiment would be more appropriate.

       Response: Thank you so much for this comment as addition of this information strengthen our data. ImageJ assisted quantification of CYP1A1 staining in neurons is shown in the table included in Figure 5 with the mean value and standard deviation indicated.

      Comment: Values (means and errors) of all experiments must be indicated in the main text;

Minor comments:

Comment: Please report scale bars on all immunofluorescence images (in Figs 1B, 5, 6 and Fig. S1). Moreover, technical details of image acquisition (objective used, magnification, image resolution in terms of pixels) are lacking and should be added.

Response: Details of objectives and magnification used have been added to the Materials and Method section.

Comment: Figs. 1E and 1G are mentioned in the text but are non reported in the figure, please check them;

Response: This has been corrected.

Comment: At line 216, authors indicated Fig 6B. Please check if they intended Figure 5

Response: Thank you for pointing this out. The figure numbering has been corrected.

       Comment:  In the figure legend of Fig 5 please specify what the arrows indicate

Response: This information has been added.

Comment:   Figs 2D-G and 5 have no reference in the text;

Response: This has been rectified. Figure 5 was not correctly referred to in the text and now replaced with 5A and 5B

Comment:   Author should justify the concentration of TCDD ( 2.5, 5 and 10 nM) of L-Kyn ( 1 and 40μM) and of KA Kynurenic Acid (1 and 40μM)

Response: thank you for this comment. The set of the two experiments were performed due to the fact that the other endogenous ligands (L-Kyn and KA) were reported previously as weak ligand of AhR and we selected the concentration on the basis of this rationale for the concentrations used has been discussed above and the reference has been included in the text. There were some initial screenings of different doses were also performed before selecting the reported doses.

Comment:   Texts contain typos: “neurones” rather than “neurons”;

Response: Spelling has been changed to ”neuron” throughout.

Comment:    Error bars are not present in all bar graphs, please check them (e.g Fig 1D, 4 B and C);

Response: They have been checked and bar charts revised

Comment: Please indicate the values of p for each asterisks in the figure legend

Response: This is now indicated.

Comment: Please indicate the “not significance” or the p value in all bar graphs.

Response: This is now indicated.

Comment: Please specify the software used for statistical analysis;

Response: This information has been included.

Comment:  In the materials and methods please indicate:  the secondary antibodies used for Immunofluorescence and the concentration of primary and secondary antibodies. In the paragraph, “Neural differentiation from “hIPSC” please reported the final concentration of all components.

Response: The primary and secondary antibodies and the dilutions used are mentioned in Table S2 in supplementary file.

Reviewer 2 Report

It is known that several ligands of the Aryl hydrocarbon receptor (AhR) are neurotoxic and therefore can contribute to the development of neuronal disorders. To analyze the impact of AhR signaling in human neurons, the authors generated iPS-derived human motor neurons and analyzed AhR-signaling in this model. This iPSC model is rather interesting and can be used to analyze not only AhR-signaling on neurons but can be useful for other signaling pathways as well. Nevertheless some issues and questions are open and need to be solved.

Major points:

Is it possible to refer to materials and methods at the beginning of the result section? Especially the start to the results section is a bit confusing when the M+M section can not be found easily. Also the description of the reprogramming and verification of this is only poorly described.

EROD measurements and CYP1a1 expression are depticed in Fig 5 not 6. MNF does not reduce EROD activity in these experiments, or to a only minor degree. How can this be explained?

Figure 3: In their data, TCDD enhances CYP1A1 mRNA in neurons, while CYP1B1 and AhRR are reduced. How can this be explained? Also iPSC downregulate CYP1a1 after TCDD treatment; is there an explanation for that?

The induction of CYP1B1 is much higher by Kyn and KA compared to TCDD. Are there other signaling pathways know to be activated by Kyn or KA to explain this? Other pathways would also be in line with the observation that MNF does not block Kyn or KA induced CYP1B1

Minor points:

Check the spelling neurons vs neurones throughout the manuscript

Check the lane spacing also throughout the manuscript

Figure 3: sublabels “C” and “E” are missing; upper and lower parts of Figure 3B are cut off, and Fig 3B is blurry; legend and figure: include the TCDD concentrations, as shown in Figure 4

Author Response

Responses to  Reviewer 2

It is known that several ligands of the Aryl hydrocarbon receptor (AhR) are neurotoxic and therefore can contribute to the development of neuronal disorders. To analyze the impact of AhR signaling in human neurons, the authors generated iPS-derived human motor neurons and analyzed AhR-signaling in this model. This iPSC model is rather interesting and can be used to analyze not only AhR-signaling on neurons but can be useful for other signaling pathways as well. Nevertheless some issues and questions are open and need to be solved.

Major points:

Comment:  Is it possible to refer to materials and methods at the beginning of the result section? Especially the start to the results section is a bit confusing when the M+M section can not be found easily. Also the description of the reprogramming and verification of this is only poorly described.

Response: Many thanks to the reviewer for this suggestion and the only reason to include the MM in supplementary was to avoid the extra length of the article. But Now the part is incorporated with the main text. The reprogramming protocol was precisely described due to its kit protocol, which is described in detail by the kit provider.

Comment:  EROD measurements and CYP1a1 expression are depicted in Fig 5 not 6. MNF does not reduce EROD activity in these experiments, or to a only minor degree. How can this be explained?

Response: The figure number has been corrected in text.

In the discussion section, we mentioned that the MNF has been reported to have a dual role depending on experimental conditions, cells and organs studied. This has already been reported by other researchers., the inhibitory capacity is affected by conditions and type of cells. In fact it would be interesting to study the antagonistic and agonistic nature of various other known molecules using this system.

Comment: Figure 3: In their data, TCDD enhances CYP1A1 mRNA in neurons, while CYP1B1 and AhRR are reduced. How can this be explained? Also iPSC downregulate CYP1a1 after TCDD treatment; is there an explanation for that?

Response: We updated the text and discussed the this point in the second paragraph of the Discussion. (From Line 482-499)

Comment: The induction of CYP1B1 is much higher by Kyn and KA compared to TCDD. Are there other signaling pathways known to be activated by Kyn or KA to explain this? Other pathways would also be in line with the observation that MNF does not block Kyn or KA induced CYP1B1

Response: Thanks for these comments. In the literature, there are several mechanism reported to induce AHR and its target genes which are involved in many pathological conditions (1)

In liver, the main function of kynurenine pathway is to produce redox cofactor nicotinamide adenine dinucleotide (NAD). Furthermore, kynurenine and 3-hydroxykynurenine are degraded to kynurenic acid and xanthurenic acid by kynurenine aminotransferase (KAT). This is proposed in various studies that, Kynurenine can act as a ligand for aryl hydrocarbon receptor (AHR). AHR functions either as a transcriptional regulator in adaptive xenobiotic response or as an immunomodulator in physiological response. In one of the reported studies (2), mRNA expressions of AHR, CYP1A1 and CYP1B1 by tryptophan or kynurenine treatment. AHR expression was significantly increased by tryptophan which resulted in the higher induction of the CYP1B1 than CYP1A1. Our study was limited to in vitro system using hiPSCs to motor neurons and unfortunately, the regulation of the AHR induced signaling to these enzyme are unknown in many of such studies.

  1. J. Mol. Sci.2021, 22(3), 1104; doi.10.3390/ijms22031104
  2. Toxicol.2020 96, 282-292 doi.10.1016/j.reprotox.2020.07.011

Minor points:

Comment: Check the spelling neurons vs neurones throughout the manuscript

Response: The spelling has been changed to xxx throughout the manuscript.

Comment: Check the lane spacing also throughout the manuscript

Response: The line spacing has been corrected

Comment: Figure 3: sublabels “C” and “E” are missing; upper and lower parts of Figure 3B are cut off, and Fig 3B is blurry; legend and figure: include the TCDD concentrations, as shown in Figure 4

Response: Unfortunately, this problem was caused by some software errors and has now been rectified .

Round 2

Reviewer 1 Report

 Accept in present form